# Differential Deformation Identification of High-Speed Railway Substructures Based on Dynamic Inspection of Longitudinal Level

**DOI:** 10.3390/s23010219

**Published:** 2022-12-25

**Authors:** Shuai Ma, Xiubo Liu, Bo Zhang, Jianmei Wei

**Affiliations:** Infrastructure Inspection Research Institute, China Academy of Railway Sciences Corporation Limited, Beijing 100081, China

**Keywords:** differential deformation, identification, substructure, high-speed railway, longitudinal level, convolutional neural network

## Abstract

High-speed railway administrations are particularly concerned about safety and comfort issues, which are sometimes threatened by the differential deformation of substructures. Existing deformation-monitoring techniques are impractical for covering the whole range of a railway line at acceptable costs. Fortunately, the information about differential substructure deformation is contained in the dynamic inspection data of longitudinal level from comprehensive inspections trains. In order to detect potential differential deformations, an identification method, combining digital filtering, a convolutional neural network and infrastructure base information, is proposed. In this method, a low-pass filter is designed to remove short-waveband components of the longitudinal level. Then, a one-dimensional convolutional neural network is constructed to serve as a feature extractor from local longitudinal-level waveforms, and a binary classifier of potential differential deformations in place of the visual judgement of humans with profound expertise. Finally, the infrastructure base information is utilized to further classify the differential deformations into several types, according to the positional distribution of the substructures. The inspection data of four typical high-speed railways are selected to train and test the method. The results show that the convolutional neural network can identify differential substructure-deformations, with the precision, recall, accuracy and *F*1 score all exceeding 98% on the test data. In addition, four types of deformation can be further classified with the support of infrastructure base information. The proposed method can be used for directly locating adverse substructure deformations, and is also becoming a promising addition to existing deformation monitoring methods.

## 1. Introduction

Nowadays, the high-speed railway (abbr. HSR) in China has become the predominant and preferred means of transport, due to the merits of efficiency and comfort. However, the comfortable and safe operation of high-speed vehicles is inevitably dependent on the stability of railway substructures, including subgrade, bridge and tunnel, as depicted in Figure 1. In addition, the deformation of substructures, especially differential deformation, is a kind of frequently occurring disease, threatening the comfort and even the safety of passengers if the deformation develops to a certain magnitude.

In general, the differential deformation of substructures has several forms of existence. The first is the subgrade differential settlement due to foundation deformation, typically induced by groundwater exploitation [1], structural self-weight and cyclic vehicle load [2]. Moreover, the subgrade frost-heave in seasonal frozen-soil regions, due to the volume expansion of aqueous soil in the natural low temperature environment [3], is another type of subgrade deformation. The second is the differential settlement of piers, which can be attributed to similar causes as the subgrade differential settlement. The third is the differential settlement in tunnels, where the deformation of the inverted arch is the predominant disease, due to the complexity of geological conditions and operation interference. The last is the differential settlement of bridge-subgrade or tunnel-subgrade transitions, induced by the difference in settlement velocity and stiffness of different structures [4]. As for the railway infrastructure containing superimposed layers of substructures and superstructures from the bottom up, the above-mentioned substructure displacement will eventually be transferred to the rail surface, leading to vertical unevenness along the railway line, i.e., the longitudinal level [5]. As a consequence, the dynamic responses of vehicles running on these sections will be aggravated [1,6], evoking comfort and safety issues. Moreover, the vehicle dynamic responses induced by substructure differential settlement will in turn accelerate the deterioration of longitudinal level irregularities [7] and railway infrastructure [8,9], increasing maintenance and renewal costs. Therefore, it is of great importance to effectively monitor and identify the differential substructure deformation. It is also worth noting that while the deflection of the catenary [10] also threatens the safe operation of HSRs, this kind of disease is not of concern here.

Up until now, many monitoring methods have been adopted for HSR substructure deformation. Traditional methods, on the basis of settlement monitoring pile, sedimentation plate, sedimentation cup, and leveling measurement [11,12,13,14], are widely used, but also blamed for the high cost, low efficiency and large manual workload. For the purpose of automatic monitoring, optical fiber sensors [15], such as FBG (fiber Bragg grating) sensor and BOTDR (Brillouin optical time domain reflectometry), are also adopted, but mainly for the settlement monitoring of the deep soil layer [11], and they are usually impractical for covering the whole range of a HSR line. In recent years, wide-area monitoring methods based on GNSS (global navigation satellite system) [16,17] and InSAR (interferometric synthetic aperture radar) [18,19,20] have attracted much attention from researchers and engineers. They share the merits of automatic monitoring, and full-time, wide coverage and remote sensing, but are mainly used for large-area deformation, with a shortage of relatively low precision.

In spite of the aforementioned monitoring methods, people tend to ignore track dynamic inspection by the comprehensive inspection trains (abbr. CITs) as an auxiliary means of monitoring substructure deformation. Nowadays, in order to guarantee the safe and stable operation of HSR lines, railway administrations and companies around the world utilize CITs to periodically inspect geometric track irregularities, which reflects the unevenness of the rail surface. The track inspection data (abbr. TID) are then compared with predefined regulatory limits to locate adverse track states. More importantly, according to the mapping relationship of deformation between substructures and rail surface [3,5,7], the deformation information of structures is contained in the TID and can be recognized, thus providing a convenient and efficient means of monitoring the deformation state and detecting potential diseases. In the literature, there are research studies concentrating on the identification of track deformation induced by mud pumping [21] and high temperature [22,23] by using TID. Moreover, the recognition of the 32 m cyclic-creep camber deformation of simply supported girders has also been fulfilled by early research using longitudinal-level inspection data [24,25]. The characteristic wavelengths of the deformation of the railway track and girders are below 10 m and cyclic 32 m, respectively. However, research focusing on the identification of substructure deformation with wavelengths above 40 m, using TID, are rarely found.

In light of the above situation, this study aims to fulfil the purpose of the differential-deformation identification of HSR substructures, mainly on the basis of dynamic inspection of longitudinal-level irregularity. Meanwhile, the base information account of railway infrastructures, which contains locations, lengths and types of substructures and curves, is also gathered and utilized. Then, a differential-deformation-identification method is proposed. In the method, a digital filter is designed to remove irrelevant information in the longitudinal level and a delicate convolutional-neural-network (abbr. CNN) is constructed to automatically identify potential substructure deformations. Finally, the locations of identified deformations are matched with the base information account, to determine the exact deformation types.

The rest of the paper is organized as follows. The fundamental principle of track dynamic inspection, on-board CITs, and basic information about TID are introduced in Section 2, while the proposed differential-deformation-identification method is detailed in Section 3. Then, a case study and evaluation of the method are presented in Section 4. Finally, Section 5 summarizes the main conclusions.

## 2. Track Dynamic Inspection

CITs are instrumented with the track-dynamic-inspection system, which mainly consists of accelerometers, displacement meters, gyroscopes, acquisition devices and signal processing computers. CITs typically conduct dynamic inspection at a commercial speed, and the major inspection item is geometric track irregularities, including longitudinal level (surface), alignment, gauge, cross-level and twist [26]. As the substructure deforms mainly in the vertical direction, it is most correlated with the geometric parameter of the longitudinal level, which is inspected in accordance with the basic theory of the inertial reference method [27], as plotted in Figure 2.

The inertial reference method builds an inertial reference in the running vehicle body by using accelerometers and gyroscopes, and measures the relative distance between the rail and the vehicle body by using displacement meters. Then, the relative position of the rail surface in the inertial coordinate, i.e., the longitudinal level, can be generated. As depicted in Figure 2, let ***Z*** represent the vertical displacement of the vehicle body, ***Y*** represent the longitudinal-level irregularity and ***W*** represent the relative distance between the rail surface and the vehicle body. In detail, ***Z*** can be calculated by the quadratic integral of vertical vehicle-body acceleration, and ***W*** is directly measured by the displacement meter. Therefore, longitudinal level ***Y*** can be calculated in accordance with the following formula:(1)$$Y=Z+W=\iint \ddot{Z}dtdt+W$$

According to the above principle, the track dynamic-inspection system synthesizes several original signals into various track-irregularity parameters, after a series of processes including A/D conversion and filtering. The TID are spatially sequential data and are discretized at 0.25 m increments along the railway track. Moreover, the measurements of longitudinal level are output in three wavebands, i.e., 1.5~42 m, 1.5~70 m and 1.5~120 m, as plotted in Figure 3. It can be seen that the amplitude of longitudinal level increases with the waveband. As the differential deformation of HSR substructures typically covers a waveband ranging from tens of meters to over one hundred meters, the longitudinal level with a waveband of 1.5~120 m is adopted.

## 3. Differential-Deformation-Identification Method

In order to identify substructure deformation and determine the deformation type, an identification method combining digital filter, CNN and a base information account, is constructed. The method takes the spatially sequential longitudinal-level inspection data as the input, and outputs the deformation type together with the corresponding position. Each component and the whole structure of the proposed method are detailed as follows.

### 3.1. Data Filtering

It is generally acknowledged that the differential deformation of railway substructures is mainly influenced by the long-wave components of longitudinal level, and the components with wavebands below 40 m are empirically considered as an interference in the identification of substructure deformation. Thus, a digital filter is designed to remove the irrelevant components of longitudinal level. In detail, a Chebyshev low-pass filter is adopted, with the pass-band cut-off wavelength of 60 m, the stop-band cut-off wavelength of 40 m, the pass-band ripple of 1dB and the stop-band attenuation of 30 dB. Furthermore, zero-phase filtering is performed by processing the longitudinal-level data in both the forward and reverse directions, in accordance with [28]. The amplitude response of the designed filter is plotted in Figure 4a, and the comparison of waveforms before and after filtering, as well as the power spectral density of the longitudinal level is also depicted, in Figure 4b,c. It can be judged that the designed filter can effectively smooth the longitudinal-level waveform, removing irrelevant disturbing components (wavebands below 40 m) and keeping those components directly related to differential substructure deformation.

### 3.2. 1DCNN Convolutional Neural Network

The longitudinal-level waveforms activated by various types of differential substructure deformation share similar characteristics of sinusoidal or cosinoidal shapes, typically with large amplitudes and wavelengths ranging from 40 m to above 100 m. Such waveform characteristics in shape are useful in the identification of differential substructure deformations. As the convolutional neural network is advantageous in learning shape properties [29], it is adopted here as a feature extractor from longitudinal-level waveforms, as well as a binary classifier of whether or not a local waveform is generated by a potential substructure deformation, with the support of two superimposed fully-connected layers. Therefore, a delicate convolutional-neural-network is designed, as depicted in Figure 5.

The inspection data of the longitudinal level is composed of amplitudes and corresponding positions. Suppose ***X*** = (*x* [1], *x* [2],…, *x* [*N*]) and ***Y*** = (*y* [1], *y* [2],…, *y* [*N*]) as the position and amplitude of the longitudinal level, respectively, and *N* denotes the data capacity. The longitudinal-level data (***X***, ***Y***) is segmented in a step-wise manner with a window length of *L*, which is set as 120 m (480 points for the discretization interval of 0.25 m), according to the upper waveband-limit of longitudinal-level inspection data. The moving step of the window can be randomly set to increase flexibility in the selection of waveform samples. Then the selected waveforms are manually labelled with 1 or 0, representing whether or not a waveform sample corresponds to a differential substructure deformation. Thus, a dataset of waveform and label pairs, i.e., {(***X_k_***, ***Y_k_***), ***y_k_***}, *k*∈[1, *K*], can be generated. In the dataset, ***X_k_*** = (*x*[*t_k_*], *x*[*t_k_* + 1],…, *x*[*t_k_* + *L* − 1]), ***Y_k_*** = (*y*[*t_k_*], *y*[*t_k_* + 1],…, *y*[*t_k_* + *L* − 1]), *t_k_*∈[1, *N* − *L* + 1], and ***y_k_*** = 1 or 0.

The amplitude ***Y*** of the longitudinal level is taken as the input of the convolutional neural network. Due to the sequential property of the input data, one-dimensional convolution as well as one-dimensional pooling is adopted in the following layers. Therefore, the proposed structure is named 1DCNN here. Four stacking layers of convolution and pooling are adopted to gradually extract key shape features in the longitudinal-level waveform samples. The extracted features are then put into two stacking fully-connected layers, to generate the most probable labels corresponding to the inputs.

In the convolutional layer, same convolution and moving stride 1 are adopted to maintain identical width between the layer input and output. Meanwhile, the size of the convolutional kernels is set as 5, and the rectified linear unit (abbr. ReLU) [30] is taken as the activation function. The mapping relationship between the input sequence and the output feature map of a convolutional layer can be interpreted using the following formula:(2)$$Y_{\left(i\right)}^{d+1}=B_{\left(i\right)}^{d}+\sum_{j=1}^{C^{d}}CK_{\left(i,j\right)}^{d}*Y_{\left(j\right)}^{d}$$
where *CK^d^*_(*i*,*j*)_ represents the convolutional kernel linking the *j*th input channel, ***Y**^d^***_(***j***)_, of the *d*th convolutional layer and the *i*th output feature map, ***Y**^d^*^+1^**_(***i***)_, which will also be the input of the *d*+1th convolutional layer; *B^d^*_(*i*)_ represents the bias; * represents the convolution operation; *C^d^* is the channel size of the *d*th convolutional layer. The convolutional kernel strides over the whole width of ***Y**^d^***_(***j***)_, conducting dot product at each stride, as shown in Figure 6. Then all the *C^d^* convolution channels are added up, together with the bias, to generate a feature map. After the convolution operation, the channel size of input ***Y**^d^*** is doubled, without altering its width.

In the pooling layer, max-pooling is adopted with kernel size 2 and stride 2. After the pooling operation, the width of feature map ***Y**^d^*^+1^** is halved, without altering the channel size.

In the fully-connected layer, the number of hidden neurons is set as 128, with a drop rate of 0.2. The output is a scalar representing the most probable label with respect to a waveform sample. Furthermore, a predicted label, ***ŷ_k_***, is classified as positive if the value is above 0.5, indicating that the longitudinal- level waveform is induced by a differential substructure deformation. The specifics of the 1DCNN structure are detailed in Table 1.

The mean square error between the predicted labels, ***ŷ***, and the actual labels, ***y,*** is taken as the loss function, with the *L*_2_-norm regularization of model parameters added, to enhance generalization. In addition, a stochastic-gradient-descent algorithm is adopted to train 1DCNN with a learning rate 0.001.

### 3.3. Classification Using Base Information

The main function of 1DCNN is to recognize potential differential substructure-deformations through extracting the shape features of local longitudinal-level waveforms and binary classification. However, the exact deformation type of each waveform is still unclear. In this section, we further classify the recognized longitudinal-level waveforms into five types, i.e., vertical curve, transition deformation, pier deformation, tunnel deformation and subgrade deformation, with the aid of the base information account of railway infrastructures. The base information account is available in the database of the permanent way management information system (i.e., the PWMIS), of China [31]. Moreover, it is worth mentioning here that longitudinal-level waveforms in the vicinity of vertical curves have similar shape features with substructure deformations. Such waveforms are attributed to the gradient change of adjacent slopes, and should be detected, but not categorized into substructure deformation.

From the base information account, the start and end position of bridges, tunnels and slopes can be extracted, thus obtaining the spatial range of different substructures, as well as the central location of transitions and vertical curves. For transitions, the section that covers a length of 200 m around the central point is empirically taken as the transition section. For a vertical curve, based on the gradients *α*_1_ and *α*_2_ of neighboring slopes and radius *R*, the length, *S,* of the vertical curve can be estimated in accordance with the following formula:(3)$$S=R\cdot (\tan\alpha_{2}-\tan\alpha_{1})$$

Let *SP* and *EP* denote the start and end position, and subscript *vc*, *tr*, *br*, *tu*, *su* denote vertical curve, transition, bridge, tunnel and subgrade, respectively. Finally, a dataset of the spatial range, {***SP***, ***EP***} = {[*SP_vc_*, *EP_vc_*], [*SP_tr_*, *EP_tr_*], [*SP_br_*, *EP_br_*], [*SP_tu_*, *EP_tu_*], [*SP_su_*, *EP_su_*]}, of the aforementioned infrastructures can be generated. For a waveform and predicted label pair {(***X_k_***, ***Y_k_***), ***ŷ_k_***}, if ***ŷ_k_*** > 0.5, *x*[*t_k_* + *L*/2] of ***X_k_*** is iteratively compared with all the elements of ***SP*** to find the nearest infrastructure type and classify ***Y_k_*** into the corresponding deformation type. This process is named position matching, and is detailed in Figure 7. During the classification process, the vertical curves are firstly extracted, due to the fact that vertical curves and substructures are spatially overlapped but belong to different infrastructure types. Moreover, vertical curves are commonly considered as normal while differential substructure deformations are structural diseases. Thus, a waveform classified as a vertical curve should not be regarded as a substructure deformation. The four types of substructure deformation can then be easily classified, either iteratively or in parallel, for the reason that there is no spatial overlap among different substructure types.

In accordance with the above introduction, the flow chart of the differential-deformation-identification method is depicted in Figure 8. The proposed method contains three major parts, that is, the filtering of longitudinal-level data, the identification of potential substructure deformations using 1DCNN, and further classification based on base information account and position mapping. Finally, four types of differential substructure deformations, as well as vertical curves, can be identified.

## 4. Case Study

### 4.1. Data Source

Four railway lines from the China HSR network are selected as the test lines, which are representative of the busiest service routes. The TID of these test lines are all inspected by CITs, and the longitudinal level with a waveband of 1.5~120 m is selected as the data source. For convenience of expression, the test lines are denoted as line 1, line 2, line 3, and line 4 and the corresponding datasets are denoted as data 1, data 2, data 3, data 4, respectively. In the segmentation process, the random moving-step of the window is utilized to extract the most likely waveforms that exactly cover the range of potential substructure deformations and comply well with human visual judgement, without producing as many redundant waveform samples as the point-wise moving step. Due to the difficulties in the in situ verification of substructure deformations, authentic labels are practically unavailable. Manual labelling is adopted here by visually observing waveforms. Three experts with rich expertise are invited for the labelling work, during which process cross-verification is adopted until consensus is reached. Therefore, the trained 1DCNN can be regarded as a predictor of probable differential substructure-deformations, in place of human visual judgement. The manually selected waveforms are labelled with 1 or 0, representing a positive or a negative sample. In order to create a balanced dataset {(***X***, ***Y***), ***y***}, negative samples of a longitudinal-level waveform are distributed between two adjacent positive samples. An intercepted section of the longitudinal-level waveform and the corresponding labels are depicted in Figure 9. In the figure, the scattered points represent the labels and are located at the center of the waveform samples.

### 4.2. Performance Evaluation of 1DCNN

The 1DCNN are trained and tested using the above dataset. In the training process, the initial ninety percent of each dataset is adopted for training and validation, and the remaining ten percent is for testing and prediction. The 1DCNN is constructed on the TensorFlow framework, with the support of two NVIDIA GeForce GTX 1080Ti GPUs (NVIDIA Corporation, Santa Clara, CA, USA).

For the evaluation of the classification ability of 1DCNN, precision, recall, accuracy and *F*1 score are adopted as the evaluation indices. Their definitions are as follows:(4)$$Precision=\frac{TP}{TP+FP}\times 100\%$$
(5)$$Recall=\frac{TP}{TP+FN}\times 100\%$$
(6)$$Accuracy=\frac{TP+TN}{TP+TN+FP+FN}\times 100\%$$
(7)$$F1=\frac{2\times Precision\times Recall}{Precision+Recall}\times 100\%$$
where *TP* is true positive, *TN* is true negative, *FP* is false positive, and *FN* is false negative. The evaluation indices of predictions made by 1DCNN on the test data are shown in Table 2. It can be concluded from Table 2 that 1DCNN has satisfactory performance as an examiner of potential substructure deformations, with the average precision, recall, accuracy and *F*1 score of four test-lines exceeding 98%.

The time consumption for prediction on the test data of the four test-lines is 2.46 s, 0.85 s, 0.89 s and 0.89 s, respectively, that is, less than 0.008 s for a waveform sample, on average.

In addition, some *FP* and *FN* samples of Line 1 are selected and plotted in Figure 10. It can be seen that the filtered longitudinal-level waveforms contained in the shadowed box of Figure 10a,b share similar shape features of a cosinoidal type, while the former is actually a negative sample and the latter a positive sample. In addition, the predictions of the two samples made by 1DCNN are 0.570 and 0.471, respectively, both approximating the classification threshold 0.5. This indicates that 1DCNN is less discriminative only in some special cases, which are also difficult to extinguish with the naked eye.

Trained 1DCNN can then be utilized for prediction. During prediction, a newly inspected TID can be processed in accordance with Figure 8, adopting point-wise segmentation without labelling. For each segmented waveform, the corresponding output of the 1DCNN can be referenced as the possibility that the waveform indicates a potential substructure deformation. In this way, the whole range of the TID can be traversed, and all the potential differential substructure-deformations can be located and detected.

### 4.3. Identification of Deformation Types

Recognized substructure deformations need to be further classified into certain deformation types, with the aid of a basic information account. Taking the test data of line 1, for example, among the 150 positive samples there are 54 samples of vertical curves, 44 samples of transition deformation, 34 samples of pier deformation, 8 samples of tunnel deformation and 10 samples of subgrade deformation. Some waveform examples of the above five types are plotted in Figure 11.

In accordance with Figure 11, different types of substructure deformations are characterized with similar shape features in the longitudinal-level waveform, i.e., sinusoidal or cosinoidal shapes. Thus, conventional analytical techniques such as feature matching or spectral analysis are usually incapable of effectively distinguishing different deformation types. However, it becomes obvious and simple with the support of infrastructure base information.

In Figure 12, four waveform samples of subgrade deformation from Line 3 are plotted. This HSR line is located in Northeast China, which is the seasonal frozen-soil region. The waveform samples are confirmed to be located in the subgrade frost-heave section. Such deformations can also be identified by using the proposed method.

### 4.4. Evolution of Substructure Deformations

The longitudinal-level inspection data is helpful in monitoring the deformation state of HSR substructures, with the support of the proposed identification method. Once a suspected differential substructure-deformation is identified, the evolution process of the deformation state, reflected in the longitudinal-level waveforms, can be manifested, as plotted in Figure 13.

Figure 13a illustrates the same sample of pier deformation as Figure 11c, and Figure 13b illustrates a sample of subgrade settlement deformation. The longitudinal-level data are selected once a year, from 2012 to 2021. It can be seen that the pier-deformation waveform in the shadowed box of Figure 13a shows no obvious development with time, while the subgrade settlement grows annually from 2012 to 2018, until the track maintenance work in 2019. Figure 13c illustrates a waveform sample of subgrade frost-heave deformation, and the longitudinal-level data are selected once a month. According to the waveforms in the shadowed box, it is apparent that the amplitude of the longitudinal level grows significantly in January, February and March, when the atmosphere temperature reaches the annual minimum compared to other months. In addition, the waveform recovers when the environment turns warm, after April. This proves that the substructure deformation information is actually contained in the longitudinal-level inspection data, and the proposed differential-substructure-deformation-identification method is promising in serving in affiliation with existing monitoring methods.

## 5. Discussions and Conclusions

The differential deformation of high-speed railway substructures is a frequently occurring disease, threatening the safe, stable and comfortable operation of high-speed vehicles. Traditional monitoring techniques are confronted with various difficulties in the effective supervision of such long and vast structures as high-speed railways. In situ monitoring devices such as sedimentation plates and optical sensors are usually placed at particular locations where the deformation is serious, and it is thus economically impossible to cover the whole range of an HSR line. Remote monitoring techniques such as GNSS and InSAR are mainly devoted to large-area deformation monitoring, being incapable of obtaining substructure deformation which covers a spatial range from tens of meters to above a hundred meters. Fortunately, the information of substructure deformation is contained in the longitudinal-level inspection data from regular dynamic inspections by comprehensive inspection trains, and by delving into inspection data, differential substructure deformation can be identified. Moreover, the easily available inspection data covers the whole range of an HSR line, thus greatly reducing the high cost of electronic- or optical-sensor installation.

Therefore, a differential deformation-identification method based on longitudinal-level irregularity, digital filtering, a convolutional neural network and infrastructure base information is proposed. In detail, a Chebyshev low-pass filter with the cutoff wavelength of 40m is designed to filter out irrelevant components of the longitudinal level. A convolutional neural network, with four stacking layers of convolution and pooling and two fully-connected layers, is then constructed to figure out potential differential substructure-deformations. Finally, infrastructure base information is utilized for the further classification of deformation types, based on a position-mapping algorithm. According to test results in the inspection data of four typical high-speed railway lines in China, the average precision, recall, accuracy and *F*1 score of the convolutional neural network are 98.2%, 98.3%, 98.2% and 98.2%, respectively, all exceeding 98%. The precision or recall on some test lines even reaches 100%. The specific deformation types can be efficiently classified in accordance with the mapping of data location and actual infrastructure positions. Furthermore, a series of inspection data from different inspection runs can also be analyzed for tracking the development path of deformation. The proposed method is helpful in identifying and locating probable differential substructure-deformations, and is thus valuable for maintenance work. Moreover, it may also be utilized as an affiliation or an alternative to existing deformation-monitoring techniques.

## Figures and Tables

**Figure 1 sensors-23-00219-f001:**
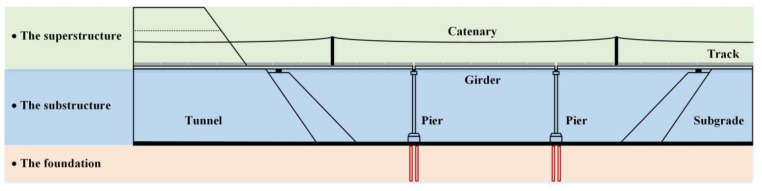
The sketch of high-speed railway structures.

**Figure 2 sensors-23-00219-f002:**
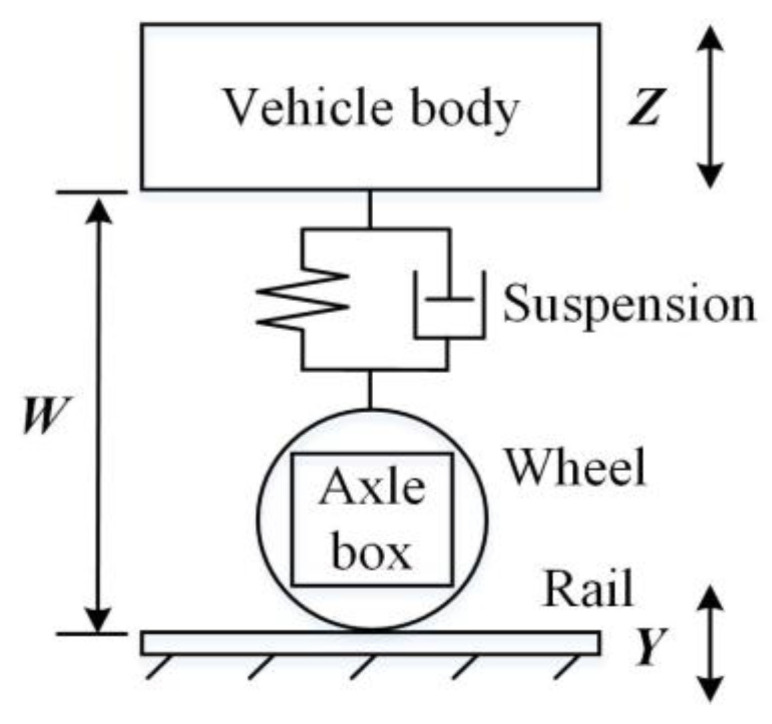
Sketch of inertial reference method.

**Figure 3 sensors-23-00219-f003:**
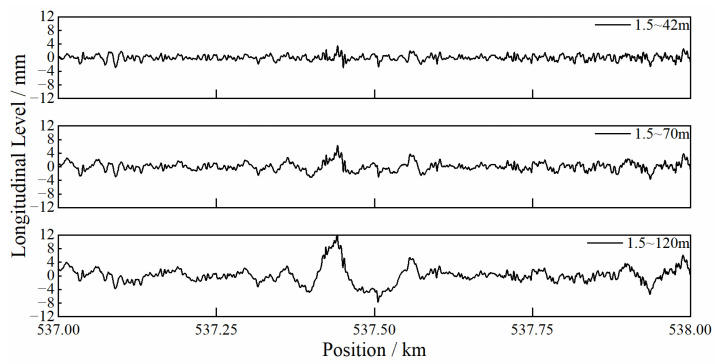
Waveform of longitudinal-level irregularity with different wavebands.

**Figure 4 sensors-23-00219-f004:**
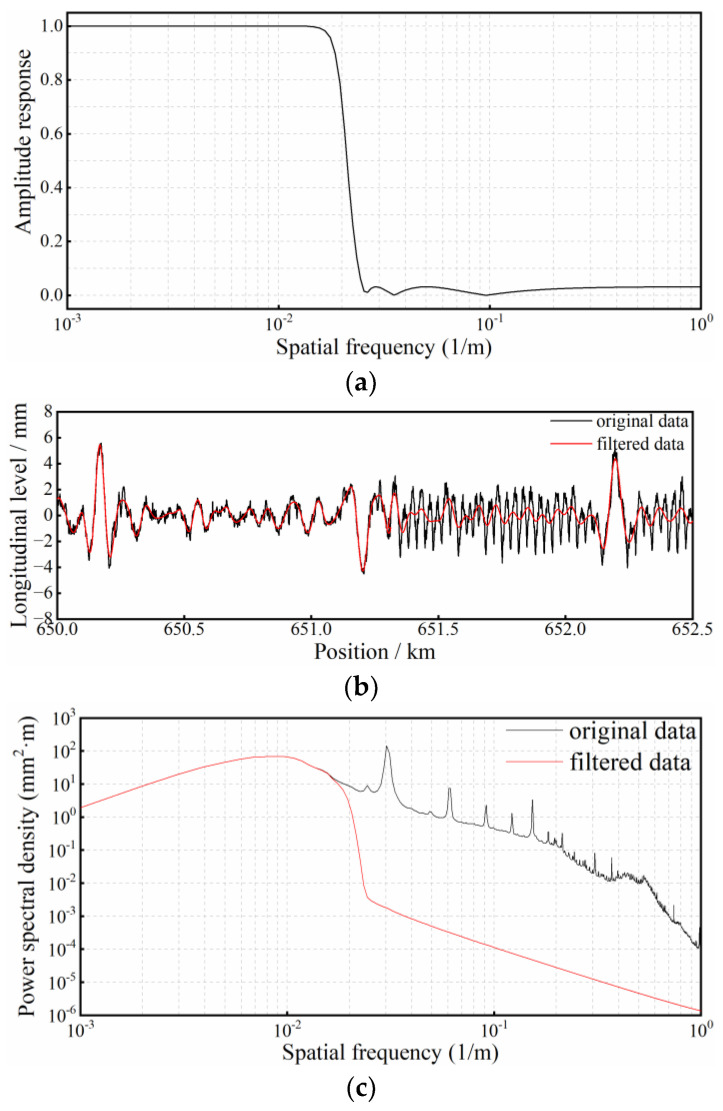
Characteristics and performance of the digital filter: (**a**) frequency response of the designed filter; (**b**) comparison of longitudinal-level waveforms before and after filtering; (**c**) comparison of power spectral density of longitudinal level before and after filtering.

**Figure 5 sensors-23-00219-f005:**
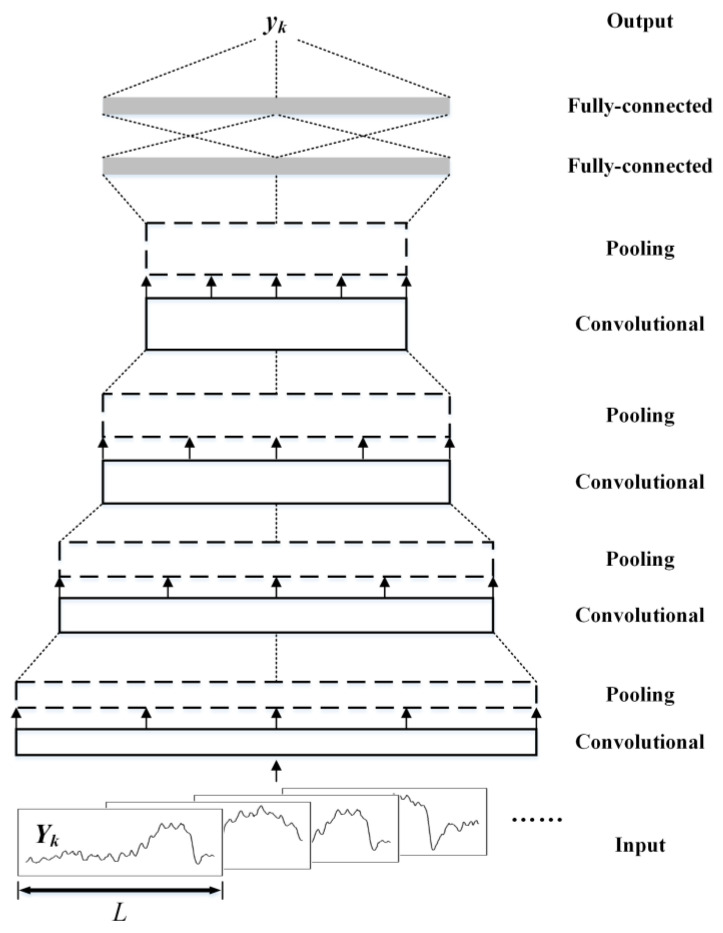
Structure of the proposed convolutional-neural-network.

**Figure 6 sensors-23-00219-f006:**
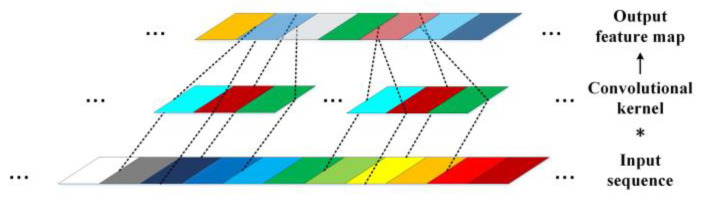
The sketch of convolution.

**Figure 7 sensors-23-00219-f007:**
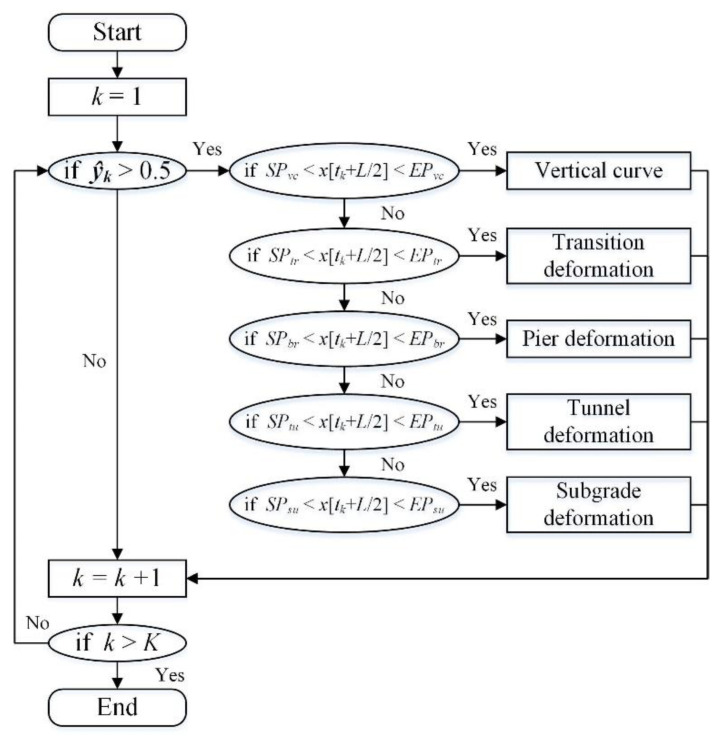
The flow chart of position-matching algorithm.

**Figure 8 sensors-23-00219-f008:**
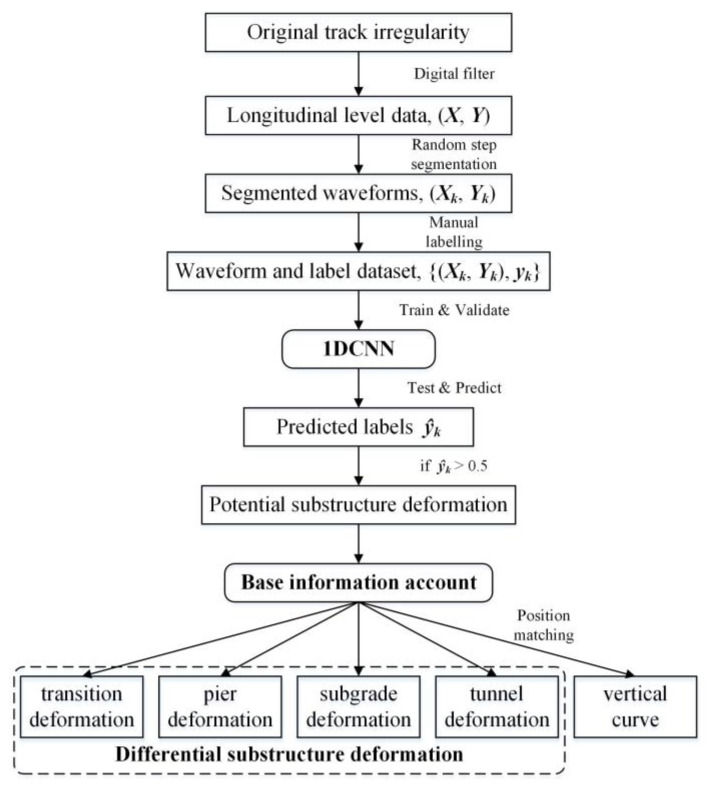
The flow chart of the differential-deformation identification method.

**Figure 9 sensors-23-00219-f009:**
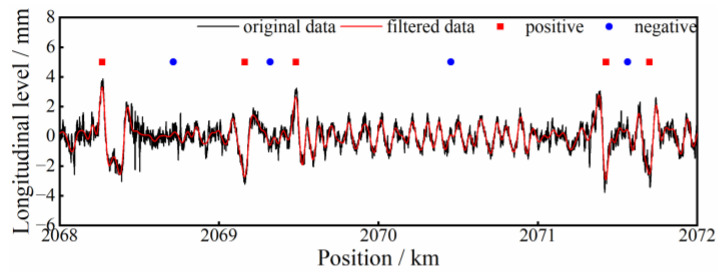
Longitudinal-level waveform and corresponding labels.

**Figure 10 sensors-23-00219-f010:**
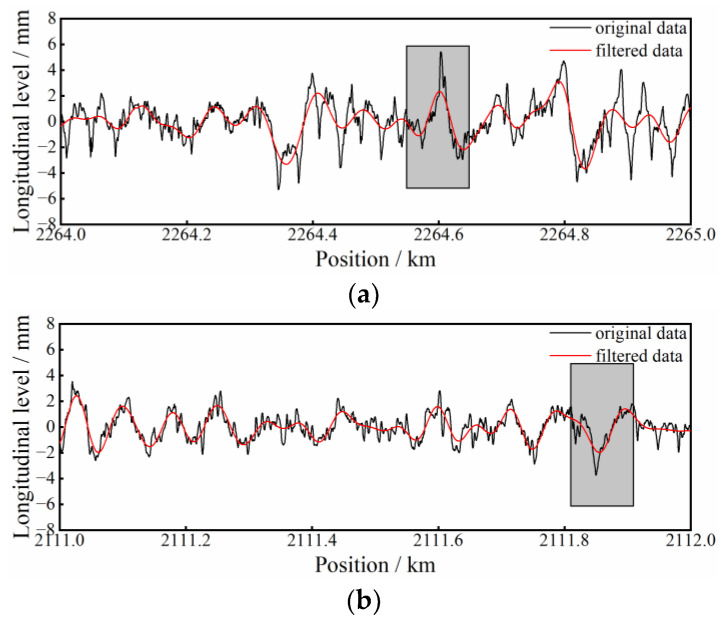
Samples of incorrectly classified waveforms: (**a**) sample of FP; (**b**) sample of FN.

**Figure 11 sensors-23-00219-f011:**
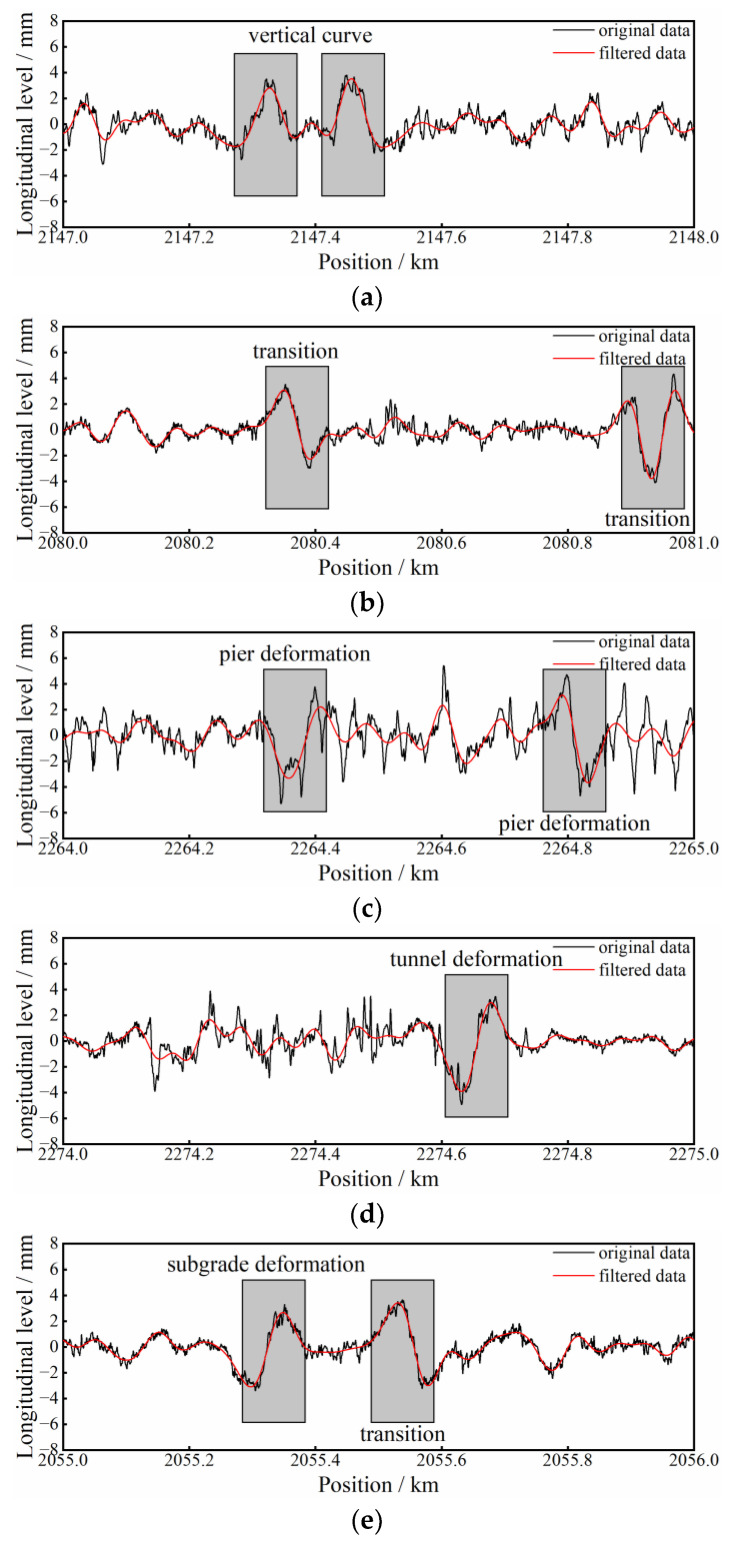
Waveform examples of different deformation types: (**a**) vertical curve; (**b**) transition; (**c**) pier deformation; (**d**) tunnel deformation; (**e**) subgrade deformation and transition.

**Figure 12 sensors-23-00219-f012:**
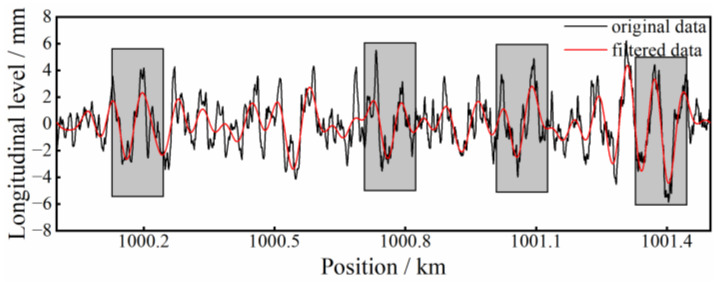
Waveform samples of subgrade frost-heave.

**Figure 13 sensors-23-00219-f013:**
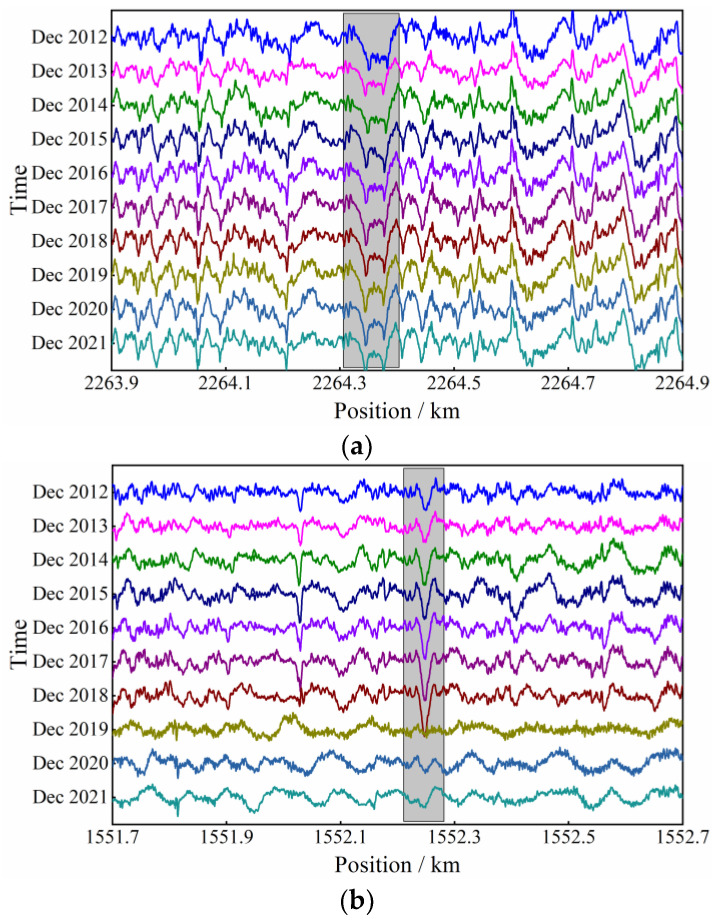
Evolution of differential substructure deformation: (**a**) pier deformation; (**b**) subgrade settlement; (**c**) frost-heave deformation of subgrade.

**Table 1 sensors-23-00219-t001:** Specifics of 1DCNN structure.

Layer	Size of Layer Output (Width × Channel)	Details
Input	480 × 1	480 continuous data points of longitudinal level
Convolutional	480 × 4	Kernel size 5, stride 1, ReLU
Max-pooling	240 × 4	Kernel size 2, stride 2
Convolutional	240 × 8	Kernel size 5, stride 1, ReLU
Max-pooling	120 × 8	Kernel size 2, stride 2
Convolutional	120 × 16	Kernel size 5, stride 1, ReLU
Max-pooling	60 × 16	Kernel size 2, stride 2
Convolutional	60 × 32	Kernel size 5, stride 1, ReLU
Max-pooling	30 × 32	Kernel size 2, stride 2
Dimension transformation	960 × 1	Reshape multi-channel features
Fully-connected	128 × 1	Dropout rate 0.2, ReLU
Fully-connected	1 × 1	Identity activation
Output	1 × 1	Scalar

**Table 2 sensors-23-00219-t002:** Statistics of evaluation indices of 1DCNN.

Line	Whole Length/km	Number of Labelled Samples (Test Data)		Number of Predicted Samples				Precision	Recall	Accuracy	*F*1 Score
		Positive	Negative	TP	TN	FP	FN				
Line 1	2294	150	147	146	146	1	4	99.3%	97.3%	98.3%	98.3%
Line 2	1318	101	100	100	95	5	1	95.2%	99.0%	97.0%	97.1%
Line 3	532	31	31	30	31	0	1	100%	96.8%	98.4%	98.4%
Line 4	964	51	50	51	49	1	0	98.1%	100%	99.0%	99.0%
Average								98.2%	98.3%	98.2%	98.2%

## Data Availability

Not applicable.

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
