# Peer review of "Differential Deformation Identification of High-Speed Railway Substructures Based on Dynamic Inspection of Longitudinal Level"

_sensors, 2022, doi:10.3390/s23010219_

Round 1
Reviewer 1 Report
In this paper, an identification method combining digital filtering, convolutional neural network and infrastructure base information is proposed, and the method is trained and tested with satisfactory results. The article is logical and clear in structure, and the research results have certain practical significance for high-speed railway maintenance. However, before publication, there are still some minor revisions that should be made.
1) In Introduction Part, It is suggested to reduce the content which is irrelevant to scientific issues, especially in the 1st paragraph.
2) In the Introduction part, the authors review a lot of studies on deformation monitoring of track structures, but rarely summarize the research on deformation identification based on track inspection data. However, the track deformation identification based on track inspection data of CIT has also been studied in some previous works, see for instance, 'Mud pumping defect detection of high-speed rail slab track based on track geometry data. Journal of Transportation Engineering, Part A: Systems.DOI: 10.1061/JTEPBS.0000678.' and 'Identification of temperature-induced deformation for HSR slab track using track geometry measurement data, Sensors 2019, 19(24), 5446; DOI: 10.3390/s19245446', which could be referred or mentioned.
3) line 111-113, the authors state that “As the substructure deforms mainly in the vertical direction, it is most correlated with the geometric parameter of longitudinal level,...”. As far as I know, the longitudinal and vertical are different directions. Please confirm the expressions.
4) In Section 3, it is suggested to write a summarized paragraph describing from where you are starting and where do you expect to arrive at the beginning of the Section.
5) In the Conclusion Part, it is recommended to indicate some statistical results.
Author Response
In this paper, an identification method combining digital filtering, convolutional neural network and infrastructure base information is proposed, and the method is trained and tested with satisfactory results. The article is logical and clear in structure, and the research results have certain practical significance for high-speed railway maintenance. However, before publication, there are still some minor revisions that should be made.
1) In Introduction Part, It is suggested to reduce the content which is irrelevant to scientific issues, especially in the 1st paragraph.
Reply: Many thanks to your pertinent suggestion. The 1st paragraph as well as other less irrelevant parts of the Introduction is shortened.
2) In the Introduction part, the authors review a lot of studies on deformation monitoring of track structures, but rarely summarize the research on deformation identification based on track inspection data. However, the track deformation identification based on track inspection data of CIT has also been studied in some previous works, see for instance, 'Mud pumping defect detection of high-speed rail slab track based on track geometry data. Journal of Transportation Engineering, Part A: Systems.DOI: 10.1061/JTEPBS.0000678.' and 'Identification of temperature-induced deformation for HSR slab track using track geometry measurement data, Sensors 2019, 19(24), 5446; DOI: 10.3390/s19245446', which could be referred or mentioned.
Reply: Many thanks to your pertinent suggestion. The authors reviewed the literature and added some relevant ones, including those papers you mentioned. These references are listed in [22-24].
3) line 111-113, the authors state that “As the substructure deforms mainly in the vertical direction, it is most correlated with the geometric parameter of longitudinal level,...”. As far as I know, the longitudinal and vertical are different directions. Please confirm the expressions.
Reply: Longitudinal level (or surface) refers to the track unevenness in the vertical direction. It is defined in the ISO standard (ISO 23054-1:2022).
4) In Section 3, it is suggested to write a summarized paragraph describing from where you are starting and where do you expect to arrive at the beginning of the Section.
Reply: Many thanks to your pertinent suggestion. A summarized paragraph is added at the beginning of Section 3.
5) In the Conclusion Part, it is recommended to indicate some statistical results.
Reply: Many thanks to your pertinent suggestion. We further detailed some statistical results and numerical conclusions in the Conclusion Part, even though such limited results mainly stay in Paragraph 4.2.
This work presents a study about the deformation identification of high-speed railway substructures based on a dynamic inspection of the longitudinal level. Generally, this paper is well-written, and the content is rich. It is recommended to be published after addressing the following issues.
1) In the abstract, it is recommended to mention the state-of-the-art monitoring technique of rail substructure. It is better to understand the shortcoming in current research and it can be an important motivation of this work.
Reply: Many thanks to your pertinent recommendation. We added explanations of the shortcoming of the existing monitoring techniques, as can be seen in the second sentence of the Abstract.
2) In the introduction, it is worthwhile to mention that apart from the substructure, the overhead infrastructure [1-2] is also of importance to the safe operation of high-speed railways, but this paper only focuses on the substructure. In this way, the introduction can give the readers a full sketch to understand the rail infrastructure.
[1] "Wind deflection analysis of railway catenary under crosswind based on nonlinear finite element model and wind tunnel test." Mechanism and Machine Theory 168 (2022): 104608.
[2] "Railway Overhead Wiring Structures in Australia: Review and Structural Assessment." Applied Sciences 12.3 (2022): 1492.
Reply: Many thanks to your pertinent suggestion. We added explanations and references at the end of Paragraph 2 in the Introduction. Moreover, Figure 1 is revised accordingly to illustrate the overhead infrastructure.
3) It is not quite understood whether the present device is mounted on a regular train or a special instrumented train. Can the train run at a commercial speed? Please clarify.
Reply: The track dynamic inspection as explained in Section 2 is installed on the comprehensive inspection train, which is an instrumented high-speed train typically running at a commercial speed. This is also supplemented in the first paragraph of Section 2.
4) Another issue is to clarify the definition of irregularity. The track irregularity has two types, namely, the static and dynamic irregularity. Please specify which one is measured by the present device.
Reply: The inspection data are collected by the track dynamic inspection installed on the comprehensive inspection train, and therefore the measurements are the dynamic irregularities.
5) What is the advantage of the present technique against existing ones? A comparative discussion should be given before claiming the effectiveness or novelty of this work.
Reply: Many thanks to your pertinent suggestion. We added discussions in the first paragraph of Section 5, and further concluded the shortcomings of existing techniques as well as the advantage of the proposed method.

Reviewer 2 Report
This work presents a study about the deformation identification of high-speed railway substructures based on a dynamic inspection of the longitudinal level. Generally, this paper is well-written, and the content is rich. It is recommended to be published after addressing the following issues.
1) In the abstract, it is recommended to mention the state-of-the-art monitoring technique of rail substructure. It is better to understand the shortcoming in current research and it can be an important motivation of this work.
2) In the introduction, it is worthwhile to mention that apart from the substructure, the overhead infrastructure [1-2] is also of importance to the safe operation of high-speed railways, but this paper only focuses on the substructure. In this way, the introduction can give the readers a full sketch to understand the rail infrastructure.
[1] "Wind deflection analysis of railway catenary under crosswind based on nonlinear finite element model and wind tunnel test." Mechanism and Machine Theory 168 (2022): 104608.
[2] "Railway Overhead Wiring Structures in Australia: Review and Structural Assessment." Applied Sciences 12.3 (2022): 1492.
3) It is not quite understood whether the present device is mounted on a regular train or a special instrumented train. Can the train run at a commercial speed? Please clarify.
4) Another issue is to clarify the definition of irregularity. The track irregularity has two types, namely, the static and dynamic irregularity. Please specify which one is measured by the present device.
5) What is the advantage of the present technique against existing ones? A comparative discussion should be given before claiming the effectiveness or novelty of this work.
Author Response
This work presents a study about the deformation identification of high-speed railway substructures based on a dynamic inspection of the longitudinal level. Generally, this paper is well-written, and the content is rich. It is recommended to be published after addressing the following issues.
1) In the abstract, it is recommended to mention the state-of-the-art monitoring technique of rail substructure. It is better to understand the shortcoming in current research and it can be an important motivation of this work.
Reply: Many thanks to your pertinent recommendation. We added explanations of the shortcoming of the existing monitoring techniques, as can be seen in the second sentence of the Abstract.
2) In the introduction, it is worthwhile to mention that apart from the substructure, the overhead infrastructure [1-2] is also of importance to the safe operation of high-speed railways, but this paper only focuses on the substructure. In this way, the introduction can give the readers a full sketch to understand the rail infrastructure.
[1] "Wind deflection analysis of railway catenary under crosswind based on nonlinear finite element model and wind tunnel test." Mechanism and Machine Theory 168 (2022): 104608.
[2] "Railway Overhead Wiring Structures in Australia: Review and Structural Assessment." Applied Sciences 12.3 (2022): 1492.
Reply: Many thanks to your pertinent suggestion. We added explanations and references at the end of Paragraph 2 in the Introduction. Moreover, Figure 1 is revised accordingly to illustrate the overhead infrastructure.
3) It is not quite understood whether the present device is mounted on a regular train or a special instrumented train. Can the train run at a commercial speed? Please clarify.
Reply: The track dynamic inspection as explained in Section 2 is installed on the comprehensive inspection train, which is an instrumented high-speed train typically running at a commercial speed. This is also supplemented in the first paragraph of Section 2.
4) Another issue is to clarify the definition of irregularity. The track irregularity has two types, namely, the static and dynamic irregularity. Please specify which one is measured by the present device.
Reply: The inspection data are collected by the track dynamic inspection installed on the comprehensive inspection train, and therefore the measurements are the dynamic irregularities.
5) What is the advantage of the present technique against existing ones? A comparative discussion should be given before claiming the effectiveness or novelty of this work.
Reply: Many thanks to your pertinent suggestion. We added discussions in the first paragraph of Section 5, and further concluded the shortcomings of existing techniques as well as the advantage of the proposed method.
Round 2
Reviewer 2 Report
All my comments have been well addressed.